

# Metabolomic and transcriptomic analyses of drought resistance mechanisms in sorghum varieties

Li Yue, Hui Wang, Qimike Shan, Zaituniguli Kuerban, Hongyan Mao and Ming Yu

Research Institute of Crops, Xingjiang Academy of Agricultural Sciences, Urumqi, Xinjiang, China

## ABSTRACT

For a long time, sorghum breeding has focused on improving yield and quality traits, whereas little research has been conducted on drought resistance. To this end, this study evaluated the phenotypes of two sorghum varieties (GL98 and GL220) under drought stress and normal conditions, and sequenced their transcriptomes and metabolomes. After drought stress, the growth rates of the roots and shoots of GL220 exceeded those of GL98 at 72 h. A total of 6,344 differentially expressed genes (DEGs) were identified *via* RNA-seq differential expression analysis; these genes were significantly annotated in the phenylpropanoid biosynthesis, starch and sucrose metabolism, amino acid metabolism, and flavonoid biosynthesis pathways. The 6,344 DEGs were clustered into four clusters by K-means, and the pathways of each cluster were annotated. A total of 3,913 metabolites were identified by ultrahigh-performance liquid chromatography–MS (UPLC–MS), and a total of 1,942 differentially accumulated metabolites (DAMs), including five common DAMs, were identified. Through combined RNA-seq and metabolomics analyses, we determined that the flavonoid biosynthesis pathway is an important regulatory pathway in the sorghum response to drought stress and that *Sobic.007G058600* was significantly correlated with 10 metabolites of the flavonoid pathway. In summary, our results provide a theoretical basis for a deeper understanding of the molecular mechanism of sorghum drought resistance and new genetic resources for subsequent research.

# INTRODUCTION

Sorghum is a resilient crop with strong drought tolerance and climate adaptability, with origins dating back over 6,000 years to northeastern Africa (*Punia et al., 2021*). Today, it is cultivated not only in Africa but also extensively in the Americas, Asia, and Europe (*Jia et al., 2021*). According to FAOSTAT data from 2020, sorghum ranks as the fifth largest cereal crop globally, following rice, wheat, corn and barley (*Girma et al., 2019*). Sorghum is a nutritious grain that contains many proteins, vitamins and minerals that help maintain health, and many fibers that help promote intestinal health (*Khalid et al., 2022*). Sorghum has versatile uses in the energy and chemical industries, including the production of alcohol, biofuels and fiber, with a variety of applications (*Nenciu et al., 2021*). Sorghum has

Corresponding author
Ming Yu, 1355258421@qq.com

broad prospects and development space for addressing global sustainable development challenges. Sorghum is recognized as a crucial crop for food security and sustainable agriculture due to its strong performance in low-input farming systems and its ability to withstand drought and heat stresses (*Pandey, Madhu & Bhat, 2019*; *Kawa et al., 2021*).

With intensifying global climate change and water scarcity, drought has become a common issue in agriculture, and its impacts on crop yield, quality, and profitability are expected to increase in the future (*Peña et al., 2020*; *Makar et al., 2022*). Therefore, enhancing crop drought tolerance is essential for ensuring high and stable crop yields (*Guadarrama-Escobar et al., 2024*). Sorghum is a drought-tolerant crop that has exceptional growth capabilities in water-limited environments, which not only makes it an important crop in agricultural production but also serves as an ideal model for studying plant responses to drought stress (*Mwamahonje et al., 2024*). Its drought tolerance originates from unique physiological and genetic characteristics. Physiologically, sorghum has a deep root system that can utilize water from deeper soil layers. Its leaves have smaller stomata and a thicker cuticle to reduce water evaporation, thereby maintaining the water balance in arid environments (*Liaqat et al., 2024*). Drought occurs when the amount of water absorbed is less than the amount of water consumed, making it difficult to meet sorghum's own water needs, thus affecting sorghum photosynthesis, protein synthesis, production and transportation of metabolites and other physiological processes and resulting in reduced sorghum yield and deterioration in quality (*Chadalavada, Kumari & Kumar, 2021*). Sorghum needs to maintain a high water content and water balance under drought stress. It can improve osmotic regulation ability by increasing the contents of proline, soluble sugars, and soluble proteins, among other substances, maintaining water retention in cells or tissues and thus maintaining turgor pressure (*Behera et al., 2022*). Under drought stress, to ensure its basic physiological functions, sorghum can also maintain its normal physiological functions through hormone regulation, increased enzyme activity, and related processes (*Liu et al., 2023*).

As sequencing technology advances quickly, molecular biology has entered the era of massive data, and multiomics sequencing technology has become a crucial tool for understanding the mechanisms of drought resistance in plants (*Roychowdhury et al., 2023*). Currently, many scholars use multiomics technology to address scientific issues related to plant responses to drought stress at the gene–protein–metabolite level to increase their understanding of plant drought resistance (*Zhang et al., 2024a*). Transcriptome and metabolome analyses are commonly employed to investigate the molecular mechanisms of plant drought stress resistance. These analyses play a crucial role in uncovering key genes, metabolites, and regulatory modules related to drought resistance mechanisms (*Li et al., 2022*). For example, the joint analysis of apple transcriptome and metabolome identified that *MdPYL9* can modulate the expression levels of the *CHS* and *CHI* genes, leading to the upregulation of the *4CL* gene. This results in the increased presence of apigenin-7-o-glucoside, enhancing drought resistance (*Liu et al., 2024*). Analysis of the transcriptome and metabolome of drought-tolerant sweet potato variety Zhenghong 23 and the sensitive variety Jinong 432 highlighted the significance of amino acid metabolism, respiratory metabolism, and antioxidant system in drought resistance

(*Yin et al., 2024*). Analysis of the transcriptome and metabolome of Salvia miltiorrhiza under varying drought conditions revealed the crucial involvement of the MAPK signaling pathway, phenylpropanoid biosynthesis, and flavonoid biosynthesis in enhancing drought resistance (*Zhou et al., 2024*). Analysis of watermelon plants under drought stress at various time points showed that 6 h is a crucial period for watermelon drought resistance. The study also identified starch and sucrose metabolism, plant hormone signal transduction, and photosynthesis pathways as key regulatory pathways for enhancing watermelon drought resistance (*Chen et al., 2024*).

Under drought stress, sorghum undergoes physiological and biochemical changes to varying degrees; these changes are affected by the expression and regulation of many drought resistance-related genes, including those related to signal transduction, to regulate gene expression at the transcriptional and translational levels. At present, some sorghum drought resistance-related genes have been identified through quantitative trait locus (QTL) mapping and homologous cloning of drought resistance genes in model plants, but there have been few reports on the regulatory mechanisms and interaction networks of genes (*Xin et al., 2022*; *Jin et al., 2023*). To this end, we performed transcriptome and metabolome sequencing on drought-resistant and drought-sensitive sorghum varieties under polyethylene glycol (PEG) stress. By analyzing differentially expressed genes and metabolites, we identified key pathways and genes associated with sorghum drought resistance. These findings offer a foundation for advancing research on the molecular mechanisms behind sorghum drought resistance and discovering new genetic resources for such studies.

## MATERIALS AND METHODS

### Drought treatment and screening of drought-tolerant materials

Plump and uniformly sized sorghum seeds were selected for the filter paper germination method. They were first washed three times with sterilized distilled water, then soaked in 1% sodium hypochlorite solution for 20 min, and rinsed 3–5 times with sterile water. The seed germination test was conducted in a constant temperature and humidity chamber at 25 °C, 60% relative humidity, 12 h of light and darkness each, and continuous culture for 72 h. Each culture dish contained 40 sterilized sorghum seeds on filter paper, with the addition of 15 mL of 20% PEG solution by mass. An equivalent amount of distilled water served as a control, and each treatment was replicated five times. Photos were taken at 12, 24, 36, 48, 60, and 72 h to measure shoot and root lengths. Samples collected at 72 h were promptly frozen in liquid nitrogen and stored at −80 °C for subsequent transcriptome and metabolome sequencing.

### Transcriptome sequencing

Total RNA from the plant was extracted following the manufacturer's protocol for the RNAprep Pure Plant Kit (Tiangen, Beijing, China). The RNA concentration, purity, and integrity were assessed with a NanoDrop 2000 and an Agilent Bioanalyzer 2100 system to verify the quality of the RNA prior to further procedures. One microgram of high-quality RNA served as the starting material for sample preparation, which we carried out using a

Hieff NGS Ultima Kit for mRNA library construction. The mRNA was purified, cDNA was synthesized, and the sequencing library was prepared following the manufacturers' recommendations, including ligation of adaptors and PCR to produce the final library, all of which were validated using a bioanalyzer. Sequencing was conducted on an Illumina NovaSeq platform, creating an extensive set of 150 bp paired-end reads. The Fastp software (version 0.23.4) was utilized to eliminate adapter sequences, filter out low-quality and N sequences exceeding a 5% ratio, resulting in clean reads for subsequent analysis (*Chen et al., 2018*). Subsequently, HISAT2 was employed to align the clean reads to the reference genome (Sorghum_bicolor.v3.1.1, https://phytozome-next.jgi.doe.gov/) (*Kim et al., 2019*). Gene function annotations were enriched and sourced from various databases, and the quantification of expression levels was accurately conducted using the fragments per kilobase of exon model per million mapped fragments (FPKM) metric.

## RNA-seq differential expression analysis

DESeq2 software (version: 1.46.0) was employed to identify differentially expressed genes (DEGs) based on gene count values in each sample, using $|$fold change$| \geq 2$ and FDR $< 0.01$ as screening criteria (*Love, Huber & Anders, 2014*). Subsequently, for all DEGs, the ClusterProfiler package (version: 4.0) in R was utilized to conduct GO and KEGG enrichment analyses *via* the hypergeometric test method (*Wu et al., 2021*). Transcription factors (TFs) were predicted and annotated utilizing the Plant Transcription Factor Database (Plant TFDB, https://planttfdb.gao-lab.org/).

## Metabolite extraction and detection

The samples were freeze-dried under vacuum using a SCIENTZ-100F lyophilization dryer (LCD Display Freeze Dryer; SCIENTZ). Following this, the dried samples were ground into a powder using an MM 400 grinding machine (30 Hz, 1.5 min; Retsch). The powdered samples were dissolved in a 70% methanol extract (1.2 mL), refrigerated overnight at 4 °C, with gentle swirling performed six times during the incubation to enhance the extraction efficiency. Subsequently, each sample was centrifuged at 12,000 revolutions per minute for 10 min, and the upper liquid fraction was filtered through a 0.22-mm membrane filter (0.22-mm pore size).

Ultra performance liquid chromatography (UPLC) conditions: The Waters Xevo G2-XS QTOF high-resolution mass spectrometer was utilized to gather primary and secondary mass spectrometry data in MSe mode using MassLynx V4.2 acquisition software (Waters). In each data acquisition cycle, dual-channel data acquisition was conducted with low and high collision energy settings. The low collision energy was maintained at 2 V, while the high collision energy ranged from 10 to 40 V. The scanning frequency for a mass spectrum was set at 0.2 s. The electrospray ionization (ESI) ion source parameters were configured as follows: capillary voltage at 2,000 V (positive ion mode) or −1,500 V (negative ion mode), cone voltage at 30 V, ion source temperature at 150 °C, desolvent gas temperature at 500 °C, backflush gas flow rate at 50 L/h, and desolventizing gas flow rate at 800 L/h.

## Metabolomics data analysis

The filtered samples were stored in containers for further analysis using ultra-performance liquid chromatography and tandem mass spectrometry. Metabolome profiling was carried out employing a broadly targeted metabolomics approach utilizing the MWDB database developed by Wuhan Mettware Biotechnology (http://www.metware.cn/). The MWDB-assigned metabolites were quantified utilizing triple-level quadrupole mass spectra acquired in multiple-reaction monitoring mode. The MWDB is a database for metabolomics analysis that can help researchers quickly and accurately analyze metabolites in different plant species; in the metabolomics analysis of sorghum, the MWDB can provide rich metabolite data. Prior to data analysis, quality control checks were carried out to verify data reliability. Principal component analysis (PCA) was utilized to investigate variabilities both between and within groups. Differential accumulated metabolites (DAMs) were assessed using orthogonal partial least-squares discriminant analysis (OPLS-DA) as described by *Kang et al. (2022)*. Metabolites with a variable importance in projection (VIP) score of ≥1 and a fold change (FC) of ≥2 (or ≤0.5) were identified and classified as DAMs.

## RESULTS

### Phenotypic evaluation of GL98 and GL220 under drought stress

To evaluate the drought resistance of sorghum varieties GL98 and GL220, we used 20% PEG to simulate drought and detected the phenotypes under stress conditions (Fig. 1A). Without PEG treatment, the roots and shoots of GL98 and GL220 were able to grow normally. The shoot lengths of GL98 and GL220 were similar, but the root length of GL98 was longer than that of GL220 (Fig. 1B). After drought stress, the lengths of the roots and shoots of GL98 and GL220 decreased, but as the treatment time increased, the growth rates of the roots and shoots of GL220 exceeded those of GL98 at 72 h (Fig. 1C). These findings indicate that GL220 has better drought resistance than does GL98 and grows more vigorously under drought stress conditions. To analyze the mechanism of the difference between GL220 and GL98 under drought stress conditions, three replicate transcriptome and five replicate metabolomics sequencing experiments were performed on roots treated with 20% PEG for 72 h and roots grown normally for 72 h.

### RNA-seq analysis

Transcriptome sequencing analysis was performed on 12 samples of roots of GL220 and GL98 under normal growth conditions and 20% PEG treatment for 72 h. After the raw data were filtered, 75.05 Gb of clean data were obtained. The proportion of Q30 bases exceeded 94.52%, and the guanine (G) and cytosine (C) content exceeded 53.73% (Table S1). First, the Pearson correlation coefficient between samples was calculated based on the fragments per kilobase of exon model per million mapped fragments (FPKM) value of each gene, and the correlation coefficient of the same sample exceeded 0.88 (Fig. 2A). Principal component analysis (PCA) clustered the replicates of the same sample together,

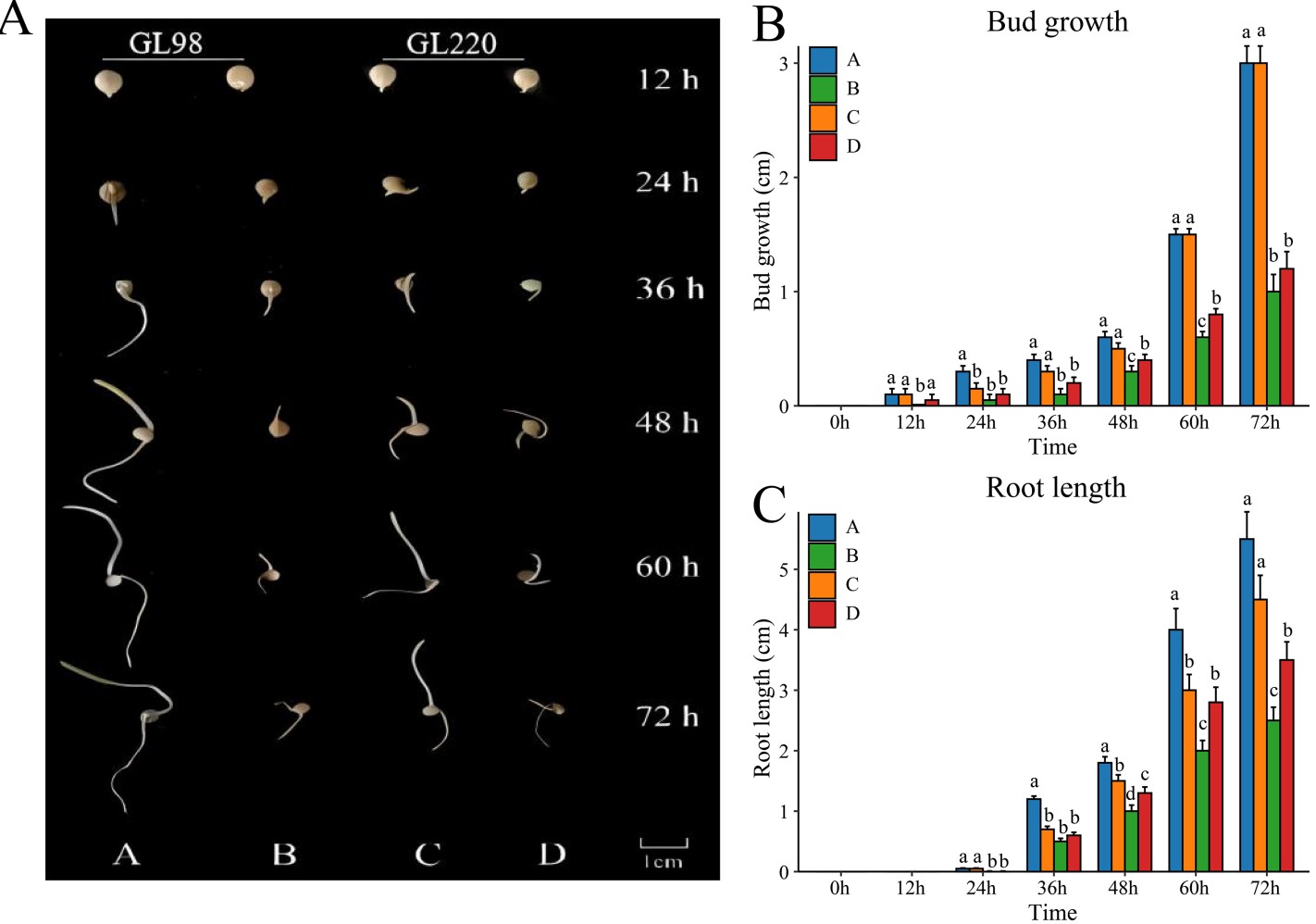

**Figure 1 Phenotypic characteristics of two different sorghum materials.** (A) Phenotypes of GL98 and GL220 under normal and drought conditions at different time points. (B) Shoot lengths of GL98 and GL220 under normal and drought conditions at different time points. (C) Root lengths of GL98 and GL220 under normal and drought conditions at different time points. The error bars represent the means ± standard deviations, the $p$ values were calculated using the t test, and different letters represent significant differences ($p < 0.05$).

whereas the samples between the treatment and the control were separated (Fig. 2B). These findings indicate that the RNA-seq data were reliable and reproducible and were suitable for further analysis.

### RNA-seq differential expression analysis

Through differential expression analysis, we identified a total of 6,344 differentially expressed genes (DEGs) (Figs. 3A and 3B). A total of 4,384 DEGs were identified between GL98 under drought stress and the control at 72 h, of which 2,682 were upregulated and 1,702 were downregulated, including 2,220 unique DEGs (Figs. 3A and 3B). A total of 3,453 DEGs were identified between the GL220 drought stress treatment group and the control group at 72 h, of which 2,435 were upregulated and 1,018 were downregulated, including 1,306 unique DEGs. A total of 724 DEGs were identified between the GL98 and
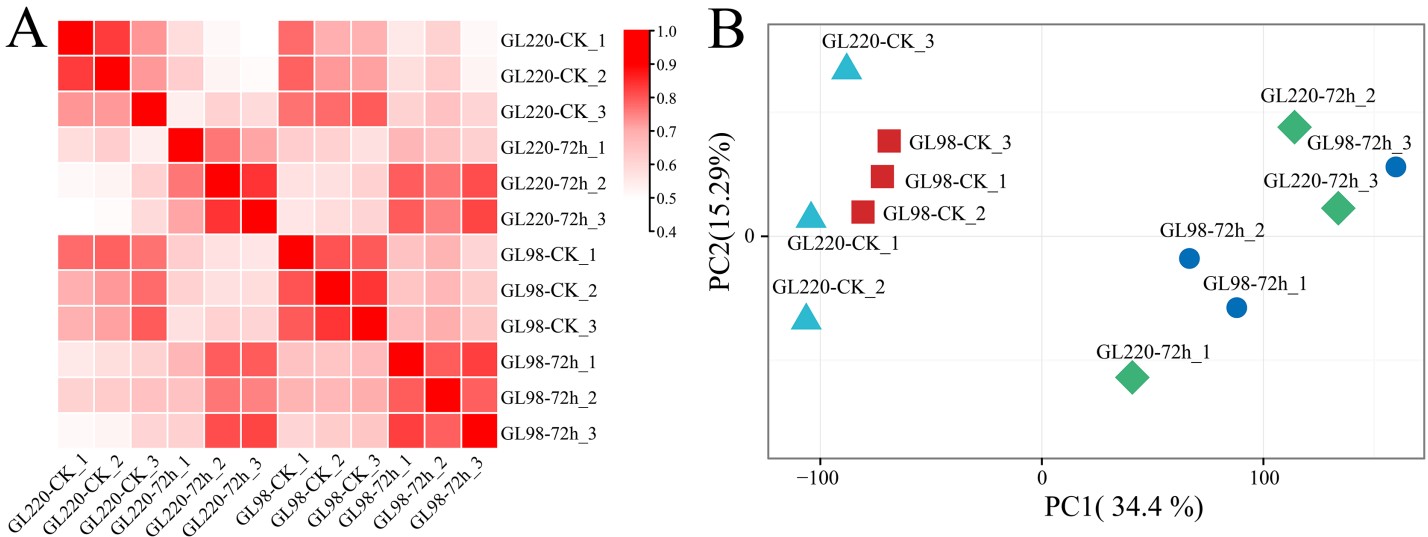

**Figure 2 Correlation and PCA of the RNA-seq data.** (A) Correlation analysis of RNA-seq samples, (B) PCA of RNA-seq samples.

GL220 controls, 319 of which were upregulated and 405 were downregulated, including 256 unique DEGs. A total of 343 DEGs were identified in GL98 and GL220 under drought stress for 72 h, of which 191 were upregulated and 152 were downregulated, including 86 unique DEGs.

To further determine the functions of the 6,344 DEGs, these DEGs were annotated *via* GO and KEGG. GO enrichment analysis revealed significant annotations related to secondary metabolite biosynthetic processes, the regulation of vesicle-mediated transport, lignin metabolic processes, lignin biosynthetic processes, phenylpropanoid biosynthetic processes, carbohydrate metabolic processes, flavonoid metabolic processes, starch metabolic processes, polysaccharide catabolic processes and the regulation of hormone levels (Fig. 3C). KEGG enrichment analysis revealed significant annotations related to galactose metabolism, transporters, carbohydrate metabolism, phenylpropanoid biosynthesis, thiamine metabolism, starch and sucrose metabolism, amino acid metabolism, fructose and mannose metabolism, flavonoid biosynthesis, cytochrome P450, carbon fixation in photosynthetic organisms and phenylalanine metabolism pathways (Fig. 3D).

## DEG clustering analysis

Genes with similar expression patterns usually have similar functions. The K-means clustering algorithm was used to cluster all the DEGs, and ultimately, four clusters were obtained (Fig. 4). The expression of Cluster 1 genes increased after drought stress in GL98 and slightly increased between the GL220 treatment and the control. A total of 437 DEGs were significantly enriched in the membrane trafficking and transporter pathways. The expression of Cluster 2 genes increased after drought stress in both GL98 and GL220. The analysis revealed 3,810 DEGs that were significantly enriched in the phenylpropanoid

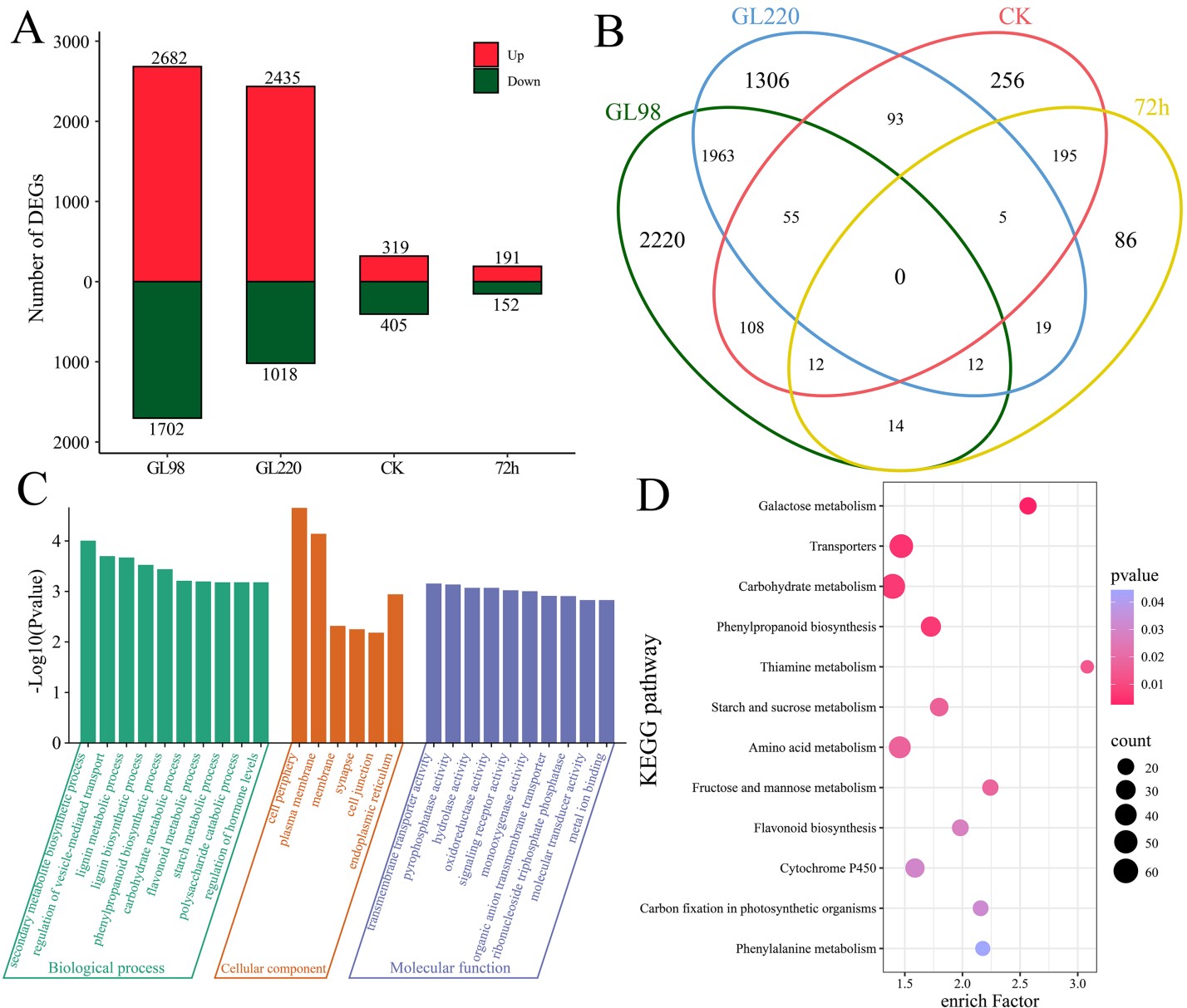

**Figure 3 Differential expression analysis and enrichment analysis of RNA-seq data.** (A) Number of upregulated and downregulated DEGs, (B) Venn diagram of the number of shared and unique DEGs, (C) GO enrichment analysis of all DEGs, (D) KEGG enrichment analysis of all DEGs.

biosynthesis and flavonoid biosynthesis pathways. The expression of Cluster 3 genes decreased after drought stress in GL98 and between the GL220 treatment and the control. There were 836 DEGs that were significantly enriched in the starch and sucrose metabolism and carbohydrate metabolism pathways. The expression of Cluster 4 genes decreased after drought stress in both GL98 and GL220. The analysis revealed 1,261 DEGs that were significantly enriched in the thiamine metabolism and fructose and mannose metabolism pathways.

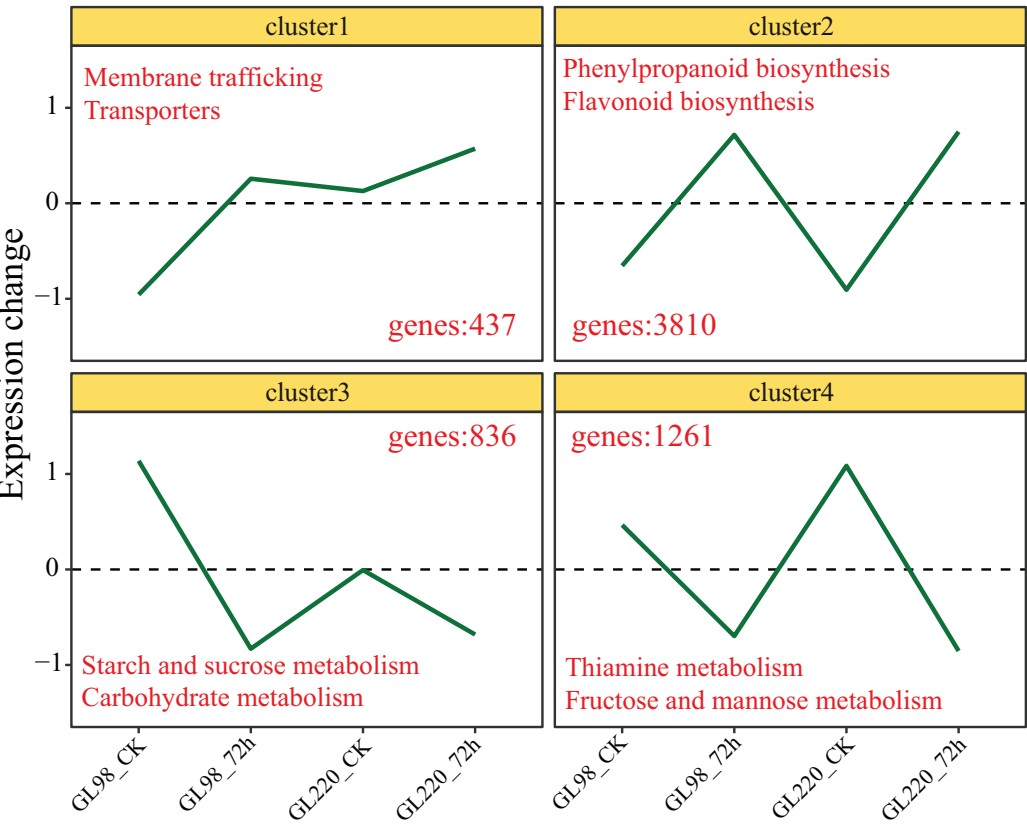

**Figure 4 The expression trend line graph of all DEGs from the cluster analysis.** The numbers in the figure represent the number of DEGs contained in different clusters, and the text represents the significantly annotated KEGG pathways.

## TF analysis

A total of 309 differentially expressed TFs were identified among the 6,344 DEGs, with AP2/ERF and MYB accounting for the greatest proportions, with 28 and 27, respectively (Fig. 5A). Hierarchical clustering was used to identify two clusters for the 309 differentially expressed TFs (Fig. 5B). Cluster 1 contained 159 TFs, and their expression levels increased in both GL98 and GL220 after drought stress. Four top TFs were identified, namely, *Sobic.003G017601* (bHLH), *Sobic.002G225100* (bZIP), *Sobic.002G189000* (HD-ZIP) and *Sobic.004G101400* (HSF). Cluster 2 contained 150 TFs, and their expression levels decreased in both GL98 and GL220 after drought stress. Four top TFs were identified, namely, *Sobic.006G080500* (bHLH), *Sobic.001G526800* (MADS), *Sobic.003G078100* (AP2/ERF) and *Sobic.003G102200* (bHLH).

## Global analysis of metabolomics data

A total of 20 samples from the four groups were detected and analyzed by UPLC-MS, which revealed 3,913 metabolites (Table S2). For example, there were 138 flavonoids, 47 steroids and 25 steroids related to plant stress resistance. These metabolites play key roles in the development of plant stress resistance. PCA revealed that the five biological

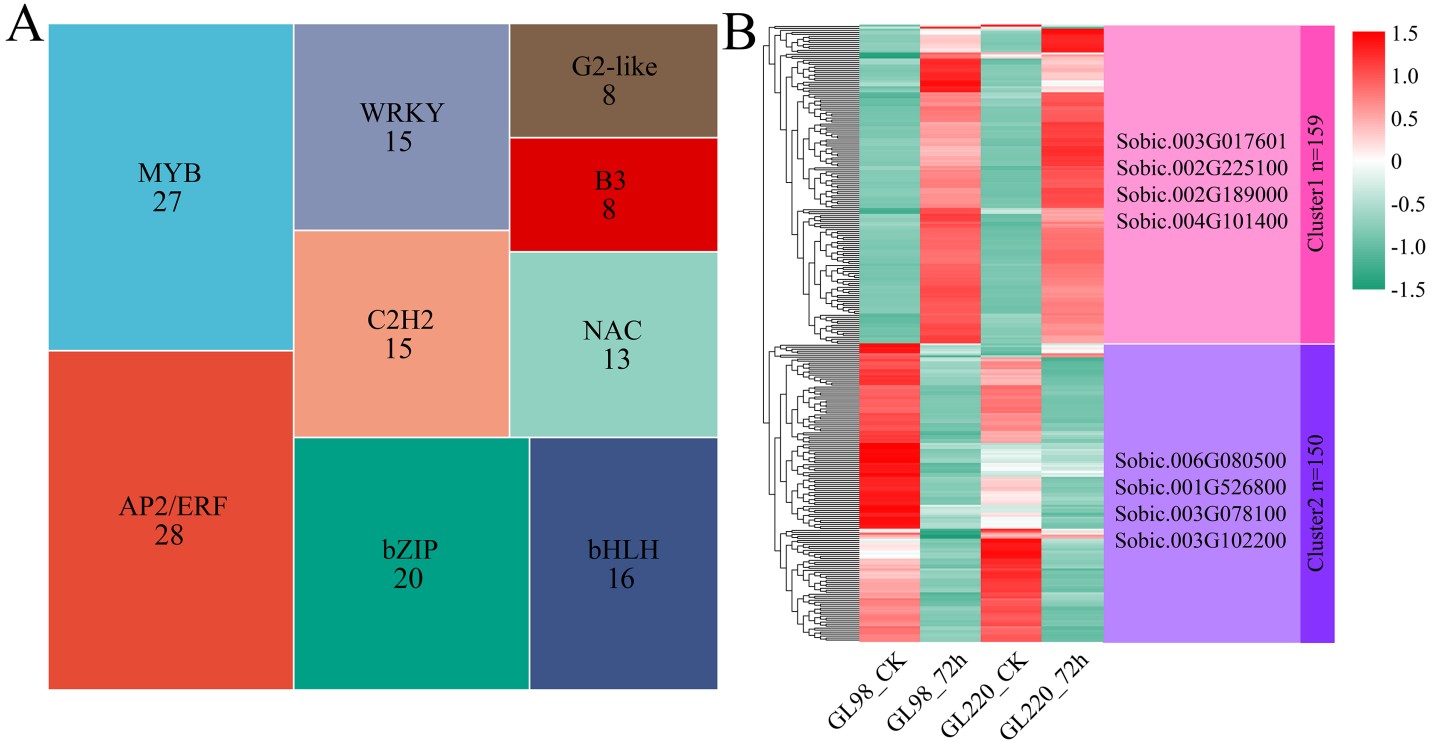

**Figure 5 Proportion of differentially expressed TFs and cluster analysis of expression patterns.** (A) Proportion of differentially expressed TFs; the numbers in the figure represent the number of transcription factors. (B) Cluster analysis of differentially expressed TFs; the right side represents the TFs with the largest fold change in each cluster.

replicates of the same treatment were clustered together, indicating that the metabolite extraction and detection techniques had good reproducibility and high reliability (Fig. 6A). After treatment, drought stress had an impact on sorghum metabolites, indicating that these metabolites changed dynamically during the response of sorghum to drought stress. The 3,913 metabolites identified were classified and annotated, and the results revealed that the metabolites were mainly divided into 10 categories: lipids and lipid-like molecules accounted for 31.20%; organic acids and derivatives accounted for 17.55%; phenylpropanoids and polyketides accounted for 12.01%; organoheterocyclic compounds accounted for 11.68%; organic oxygen compounds accounted for 10.33%; benzenoids accounted for 7.77%; nucleosides, nucleotides, and analogs accounted for 5.22%; alkaloids and derivatives accounted for 2.17%; organic nitrogen compounds accounted for 1.58%; and lignans, neolignans and related compounds accounted for 0.49% (Fig. 6B).

## DAM analysis

DAMs between samples were identified by differential expression analysis, which revealed a total of 1,942 DAMs, including five common DAMs (Figs. 7A and 7B). A total of 1,363 DAMs were identified in the GL98 drought stress and control groups at 72 h, of which 347 were upregulated and 1,016 were downregulated, including 342 unique DAMs (Figs. 7A and 7B). A total of 1,411 DAMs were identified in the GL220 drought stress and control

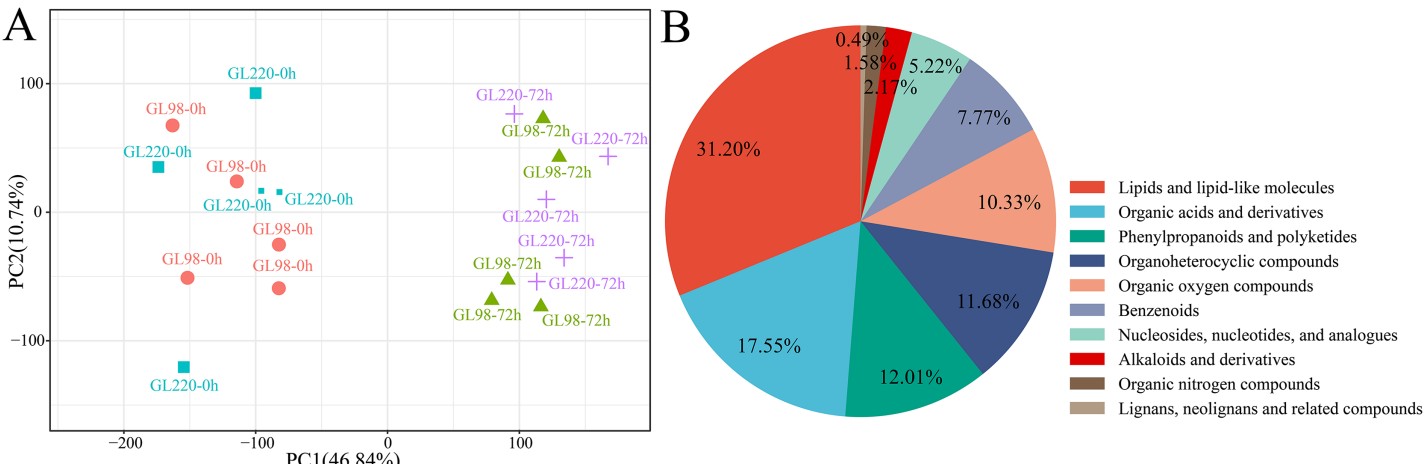

**Figure 6 Metabolome PCA and categories included in all the metabolites.** (A) PCA of different samples in the metabolome; (B) pie chart of the proportion of categories included in all metabolites.

groups after 72 h, of which 389 were upregulated and 1,022 were downregulated, including 370 unique DAMs. A total of 152 DAMs were identified between GL98 and GL220 controls, of which 68 were upregulated and 84 were downregulated, including 49 unique DAMs. A total of 232 DAMs were identified in GL98 and GL220 under drought stress for 72 h, of which 87 were upregulated and 146 were downregulated, including 71 unique DAMs.

KEGG enrichment analysis revealed that 1,942 DAMs were significantly enriched in linoleic acid metabolism, steroid biosynthesis, alpha-linolenic acid metabolism, anthocyanin biosynthesis, carotenoid biosynthesis, starch and sucrose metabolism, biosynthesis of unsaturated fatty acids, biosynthesis of various plant secondary metabolites, fructose and mannose metabolism, flavonoid biosynthesis, phenylpropanoid biosynthesis, and arginine and proline metabolism pathways (Fig. 8A). All the DAMs were clustered *via* the K-means clustering algorithm, and four clusters were ultimately obtained (Fig. 8B). The expression of Cluster 1, which included 1,030 DAMs, decreased after drought stress in both GL98 and GL220. The expression of Cluster 2, which included 338 DAMs, decreased after drought stress in GL98, with a smaller decrease in expression detected between the GL220 treatment and the control. In both GL98 and GL220, Cluster 3, which included 426 DAMs, presented increased expression after drought stress. In GL220, Cluster 2, which included 148 DAMs, presented increased expression after drought stress and a smaller increase in expression between the GL98 treatment and the control.

## Combined transcriptomics and metabolomics analyses of the drought tolerance mechanism of sorghum

Analysis of the DAMs indicated significant enrichment in pathways associated with metabolites such as phenylpropanoid biosynthesis, flavonoid biosynthesis, and anthocyanin biosynthesis post-treatment, highlighting the crucial roles of flavonoids in enhancing plant drought resistance. First, the expression patterns of 20 DEGs related to

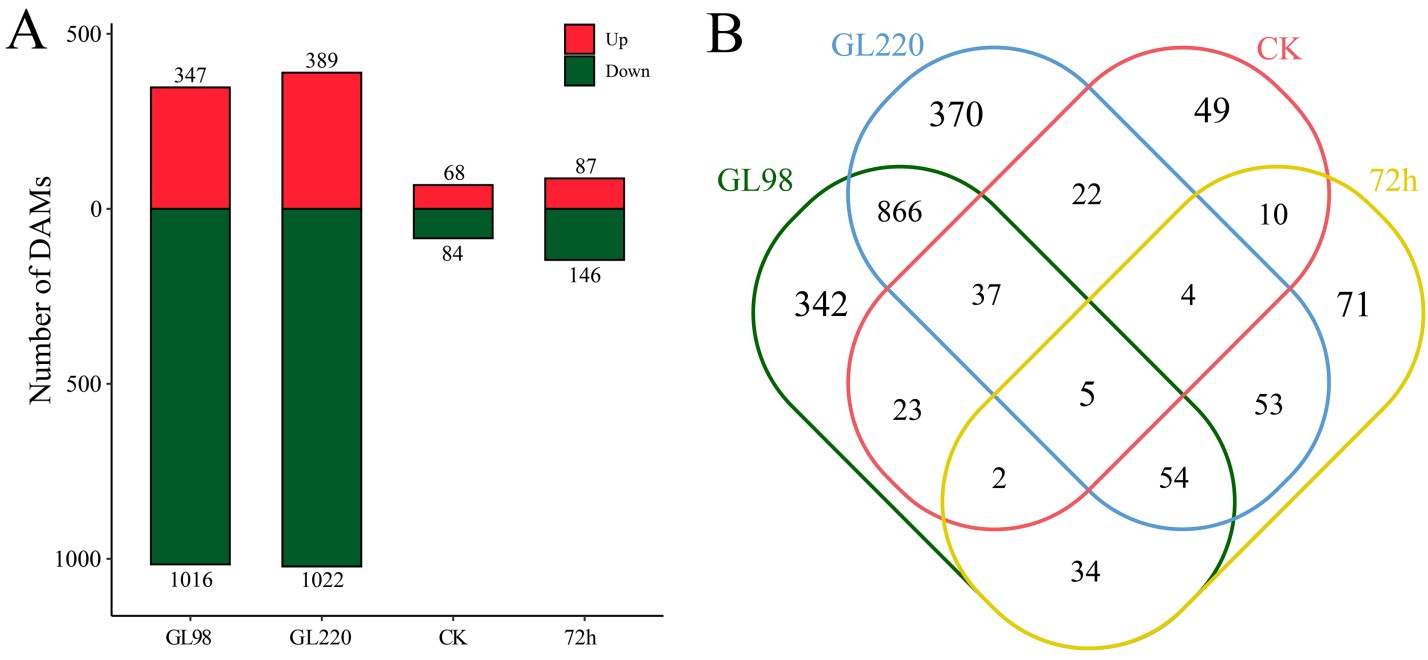

**Figure 7 Metabolomic differential analysis.** (A) Number of upregulated and downregulated DAMs. (B) Venn diagram of the number of shared and unique DAMs.

flavonoid biosynthesis were examined. Following drought stress, the majority of DEGs associated with flavonoid biosynthesis exhibited increased expression levels, with GL98 showing higher expression levels compared to GL220 (Fig. 9A). Thirteen different flavonoid metabolites were detected, but the contents of most of them decreased after drought stress. Only the contents of 5-O-caffeoylshikimic acid, pinostrobin, luteoforol, bracteatin 6-O-glucoside, and p-coumaroyl quinic acid increased after drought stress (Fig. 9B). Among them, the contents of liquorol, bracteatin 6-O-glucoside and p-coumaroyl quinic acid increased in both GL98 and GL220 after drought stress. After drought stress, the content of 5-O-caffeoylshikimic acid increased in GL98 and decreased in GL220. The pinostrobin content remained unchanged in GL98 but increased in GL220. The Pearson correlation coefficients between the DEGs of the flavonoid pathway and DAMs were computed. Genes and metabolites with absolute correlation coefficients exceeding 0.9 and $p$ values below 0.05 were selected for visualization (Fig. 9C). A total of 17 flavonoid biosynthesis genes were significantly correlated with 10 flavonoid metabolites, of which 17 genes were significantly negatively correlated with seven flavonoid metabolites and 15 genes were significantly positively correlated with three flavonoid metabolites. Among them, *Sobic.007G058600* was significantly correlated with the metabolites of the 10 most abundant flavonoid pathways, and *Sobic.010G052200* was significantly negatively correlated with only Apiforol.

## DISCUSSION

Due to escalating global warming and water scarcity, drought has evolved into a persistent worldwide issue that hampers agricultural productivity and advancement

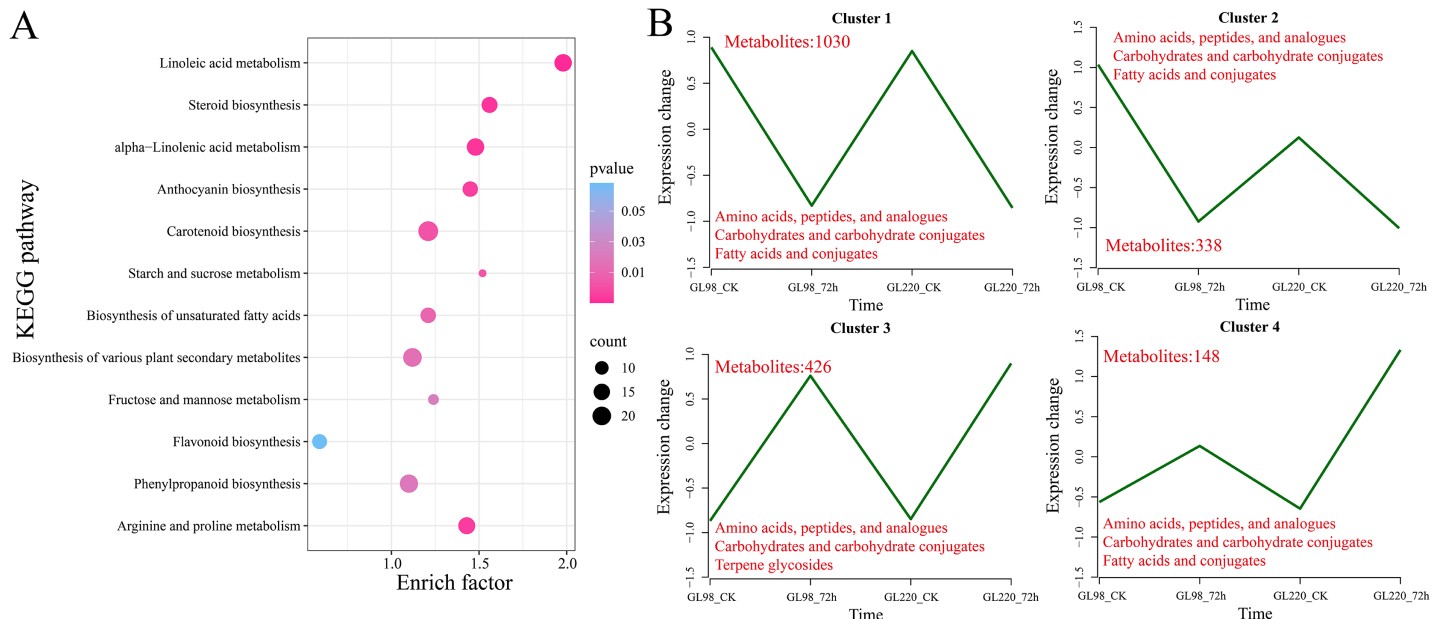

**Figure 8 DAM enrichment and clustering analysis.** (A) KEGG enrichment analysis of all the DAMs. (B) Line graph of the clustering analysis of all the DAMs, with different numbers representing the number of DAMs contained in each cluster.

(*Mourad et al., 2025*). Water scarcity significantly affects the yield and quality of essential crops, with drought-induced deficits leading to substantial growth impediments and yield reductions (*Xia et al., 2018*). Under water stress conditions, plants undergo a complex array of morphological, physiological, biochemical and molecular adaptations (*Altangerel et al., 2021*). Drought stress induces alterations in the axial symmetry development of plant leaves, resulting in leaf curling, reduced illuminated surface area, and diminished photosynthetic efficiency (*Wang et al., 2019*). Additionally, drought can lead to water loss in plant cells, impeding cell expansion and reducing turgor pressure, leading to stomatal closure, degradation of photosynthetic pigments, decreased enzyme activity in photosynthesis, leaf wilting, and diminished leaf area (*Cornic & Massacci, 1996*). This in turn leads to decreases in various attributes, such as plant height, ear length, chlorophyll levels, and root and stem biomass, ultimately resulting in a reduction in grain yield (*Anjum et al., 2011*). However, the relationships between the transcriptional and metabolic mechanisms involved in drought resistance during sorghum germination have not been studied in depth. To this end, this study used PEG (20%) to simulate drought treatment and evaluate the drought resistance phenotypes of different sorghum varieties (GL98 and GL220). Compared with GL98, GL220 has longer roots and shoots under drought stress conditions, and RNA-seq and metabolome sequencing were further performed on samples subjected to drought stress for 72 h.

One of the end products of photosynthesis in most green plants is sucrose (*Sharkey, 2024*). Sucrose, the primary soluble sugar, serves as a critical component in plant growth and transportation. Amid drought stress, nonstructural sugars, crucial in sucrose metabolism, become significant (*Zaman et al., 2025*). Functioning as an osmotic regulator,
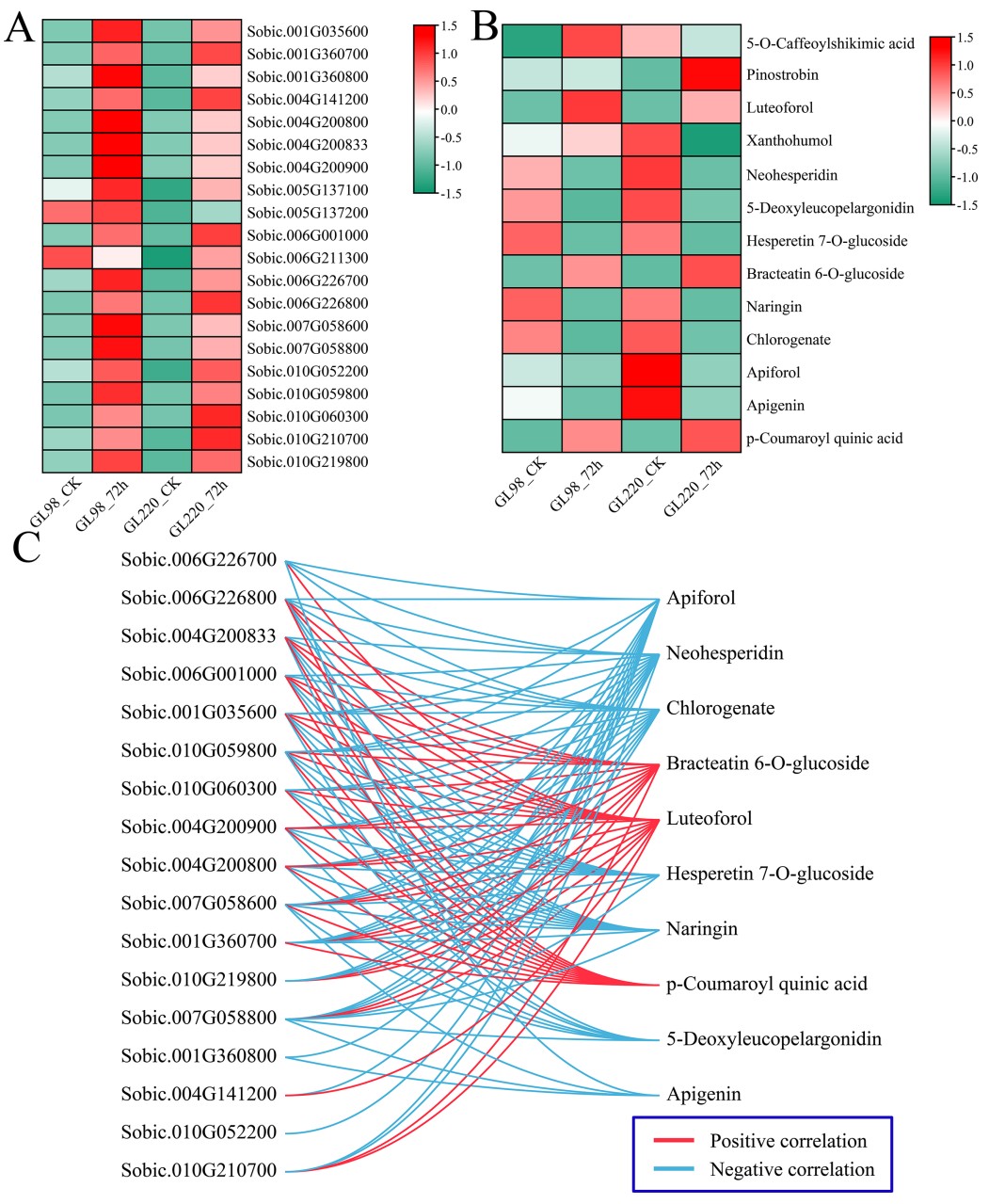

**Figure 9 Change patterns and correlation analysis of DRMs and DEGs in the flavonoid biosynthesis pathway.** (A) Changes in the expression of DEGs in the flavonoid biosynthesis pathway. (B) Heatmap of the change patterns of DRMs in the flavonoid biosynthesis pathway. (C) Correlation network diagram between metabolites and genes in the flavonoid biosynthesis pathway; blue lines represent negative correlations, and red lines represent positive correlations.

sucrose boosts the osmotic potential of plant cells. It also operates as a signaling molecule, influencing transport proteins and triggering the expression of resistance genes in plants (*Cao et al., 2024*). Moreover, sucrose can serve as an antioxidant. In situations where plant photosynthesis might be constrained by drought stress, plants typically convert starch to regenerate energy and carbon (*Afzal, Chaudhary & Singh, 2021*). The sugars and

metabolites released assist in supporting plant growth under stress, acting as osmotic protectants and compatible solutes to mitigate stress-induced adverse effects. Additionally, sugars can function as signal molecules that activate downstream components in the stress response cascade by traversing ABA-dependent signaling pathways (*Asim et al., 2023*). Moreover, in the guard cells surrounding stomata that regulate the exchange of water and $CO_2$, starch is rapidly degraded in light conditions. This degradation facilitates the production of organic acids and sugars, enhancing the expansion of guard cells and facilitating stomatal opening (*Dang et al., 2024*). Sucrose-phosphatase (SPP) is an important rate-limiting enzyme in sucrose synthesis. After drought stress, the expression level of the sorghum SPP (*Sobic.009G040900*) gene increased (log2FC = 3.07), indicating that it is an important candidate gene for improving sorghum drought resistance *via* the sugar metabolism pathway.

Transcription factors (TFs) serve as critical regulators of gene expression alterations and responses to environmental stresses, playing essential roles in plant growth, development, and environmental stress adaptation (*Rabeh, Hnini & Oubohssaine, 2025*). Major transcription factor families, including MYB, AP2/ERF, bZIP, WRKY, and HSF families, participate in modulating plant responses to diverse stresses. Overexpression of *ChbHLH93* in Arabidopsis leads to notably elevated expression levels of drought-responsive genes, enhancing the drought resistance of the transgenic plants (*Dong et al., 2025*). The overexpression of *StbHLH47* in potato resulted in greater sensitivity to drought stress, and further RNA-seq analysis revealed that many DEGs were enriched in the phenylalanine and abscisic acid signaling pathways (*Wang et al., 2025*). *MhERF113*-like overexpression resulted in increased drought resistance in apple, and ectopic expression in tomato improved drought resistance (*Tian et al., 2025*). The apple AP2/ERF transcription factor *MdDREB2A* interacts with the DRE cis-element of the *MdNIR1* promoter to regulate nitrogen utilization and sucrose transport, thereby improving the drought resistance of transgenic plants (*Zhang et al., 2024b*). *CmbZIP9* overexpression enhances the drought resistance of transgenic tobacco (*Wang et al., 2024*). We also identified 309 transcription factors that were differentially expressed under drought stress in sorghum, among which *Sobic.003G017601* (bHLH), *Sobic.002G225100* (bZIP), *Sobic.006G080500* (bHLH), *Sobic.003G078100* (AP2/ERF) and *Sobic.003G102200* (bHLH) presented log2FoldChange absolute values greater than 5. These TF families have been widely studied in the context of drought resistance in other plants. These TFs with the greatest fold changes are undoubtedly important candidate genes for future sorghum drought resistance; however, their specific functions and mechanisms still need further experimental proof.

Flavonoids, secondary metabolites prevalent in plants, play essential roles in plant development and responses to various abiotic and biotic stresses (*Shomali et al., 2022*). Producible through the phenylpropanoid metabolic pathway, flavonoids are significant secondary metabolites in plants containing antioxidant properties. Under unfavorable stress conditions, plants exhibit an elevation in flavonoid levels to a certain extent (*Ahmed et al., 2021*). Research has indicated that under drought stress, plants accumulate substantial amounts of flavonoid components. Among these, the upregulation of genes

associated with flavonoid synthesis stands out as a key mechanism by which plants respond to drought-induced stress (*Kubra et al., 2021*). The overexpression of *MsC3H29* in alfalfa can increase the primary root length and fresh weight of hairy roots under drought stress conditions. Targeted metabolomics analysis revealed that overexpression of *MsC3H29* led to increased accumulation levels of flavonoid pathway substances before and after drought stress (*Dong et al., 2024*). Compared with wild-type plants, Arabidopsis plants overexpressing *TrNAC002* presented larger leaves, increased lateral root growth, longer survival time under natural drought stress, and higher flavonoid content (*Zhang et al., 2024c*). In rice, overexpression of *OsCHI3* increased tolerance to drought stress, whereas *OsCHI3* knockout reduced rice drought tolerance. Further studies revealed that *OsCHI3* improves rice drought resistance by regulating flavonoid compounds to scavenge reactive oxygen species (*Liu et al., 2025*). Integrating transcriptome and metabolome analyses, we observed significant enrichment in the phenylpropanoid biosynthesis, flavonoid biosynthesis, and anthocyanin biosynthesis pathways. Furthermore, it was noted that the majority of genes within the flavonoid biosynthesis pathway exhibited upregulation following exposure to drought stress. The fold changes in key genes involved in flavonoid biosynthesis, such as *anthocyanidin reductase* (ANR) (*Sobic.006G226700* and *Sobic.010G210700*), *CHS* (*Sobic.005G137100* and *Sobic.005G137200*) and *CYP73A100* (*Sobic.004G141200*), were greater than 4. *O-methyltransferase* (*Sobic.007G058600*) was significantly correlated with metabolites of 10 flavonoid pathways. In summary, through RNA-seq and metabolome analyses under sorghum drought stress conditions, we determined that the flavonoid pathway plays an important role in this process, laying the foundation for further research on the molecular mechanism of sorghum drought resistance.

## CONCLUSIONS

Based on the phenotypic evaluation of sorghum under drought stress and the joint analysis of RNA-seq and metabolomics, this study revealed the molecular mechanism by which sorghum copes with alkaline stress. A total of 6,344 DEGs and 1,942 DAMs were identified. The joint analysis of RNA-seq and metabolomics revealed that the flavonoid biosynthesis pathway is an important regulatory pathway for sorghum in response to drought stress, especially genes in the flavonoid biosynthesis pathway, whose expression levels increase after drought stress. By constructing a regulatory network of flavonoid biosynthesis pathway genes and metabolites, we found that *Sobic.007G058600* was significantly correlated with a variety of flavonoid metabolites, which may play important roles in sorghum drought resistance. In summary, in this study, the molecular mechanism by which sorghum copes with drought stress was explored through multiomic analysis, and several key genes and metabolites were identified. These findings provide a theoretical basis for future sorghum drought resistance breeding.

## ACKNOWLEDGEMENTS

We thank the staff of Wuhan Metware Biotechnology Co., Ltd. (Wuhan, China), for their support during the metabolite data analysis.

## Funding

This research was funded by Xinjiang Uygur Autonomous Region Public Welfare Research Institute Basic Research Business Fee Special Project (KY2022008), Xinjiang Academy of Agricultural Sciences Youth Science and Technology Talents Project (xjnkq-2022012) and Xinjiang Minority Science and Technology Talents Project (2022D03008). The funders had no role in study design, data collection and analysis, decision to publish, or preparation of the manuscript.

## Grant Disclosures

The following grant information was disclosed by the authors:
Xinjiang Uygur Autonomous Region Public Welfare Research Institute Basic Research Business Fee Special Project: KY2022008.
Xinjiang Academy of Agricultural Sciences Youth Science and Technology Talents Project: xjnkq-2022012.
Xinjiang Minority Science and Technology Talents Project: 2022D03008.

## Competing Interests

The authors declare that they have no competing interests.

## Author Contributions

- Li Yue conceived and designed the experiments, prepared figures and/or tables, authored or reviewed drafts of the article, and approved the final draft.
- Hui Wang performed the experiments, prepared figures and/or tables, and approved the final draft.
- Qimike Shan performed the experiments, prepared figures and/or tables, and approved the final draft.
- Zaituniguli Kuerban analyzed the data, authored or reviewed drafts of the article, and approved the final draft.
- Hongyan Mao analyzed the data, authored or reviewed drafts of the article, and approved the final draft.
- Ming Yu conceived and designed the experiments, authored or reviewed drafts of the article, and approved the final draft.

## DNA Deposition

The following information was supplied regarding the deposition of DNA sequences:
PRJNA1180449

## Data Availability

The RNA-seq data of the rhizomes of the two materials is available at NCBI: PRJNA1180449.
The metabolomics data is available in MetaboLights: MTBLS11479.

## Supplemental Information

Supplemental information for this article can be found online at http://dx.doi.org/10.7717/peerj.19596#supplemental-information.

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
