# Peer review of "Metabolomic and transcriptomic analyses of drought resistance mechanisms in sorghum varieties"

_PeerJ, doi:10.7717/peerj.19596_

## Round 0.1 · original submission · Major Revisions

After you have completed the major revisions, your manuscript will be re-evaluated.

Reviewer 1 ·

Basic reporting

This manuscript provides the physiological, metabolic, and molecular effects of drought stress in sorghum by combining metabolomics and transcriptomics approaches. The study provides useful information regarding the change in metabolite and gene expression with special emphases on flavonoid biosynthesis pathway, the level of transcription factors, and some of the cluster of metabolites. Nonetheless, the study contributes significantly to the understanding of plant stress physiology, but there are some points that need to be clarified and improved to strengthen the manuscript’s scientific and logical basis.
In introduction section, highlight sorghum’s potential for both food and industrial uses in addressing global sustainability challenges.
While the need for enhancing drought tolerance is clear, the rationale lacks specificity regarding the choice of sorghum as the study focus compared to other cereals. Emphasize on how sorghum’s genetic diversity and adaptability make it uniquely suited for drought research.
Condense the gene-specific information and focus on how transcriptomics has advanced the understanding of drought resistance mechanisms in sorghum. Highlight the gaps in knowledge that the current study aims to address.
Elaborate on how metabolomics complements transcriptomics in identifying key pathways and biomarkers, creating a cohesive rationale for the integrated approach used in this study.
Highlight the novelty of this study in terms of methodology or scope compared to prior research.
In material and methods, specify how long the seeds were exposed to the 20% PEG solution and under what environmental conditions (e.g., temperature, light/dark cycle). Include details such as incubation time, temperature, and light conditions during the germination process.
The rationale for selecting 20% PEG as the drought stress inducer is not provided. Cite references or explain why this specific concentration was chosen to simulate drought stress.
The description of metabolomics analysis lacks key technical parameters such as chromatographic conditions (e.g., column type, flow rate, and solvent system) and the mass spectrometer settings. Provide specific details of the chromatography and mass spectrometry setup.
Mention the relevance and appropriateness of MWDB for sorghum-specific metabolomic analysis.
Specify the tissue type, developmental stage, and conditions under which samples were collected for RNA extraction.
Provide the source and version of the reference genome and key parameters used in the mapping process.
Explain why transcription factor analysis includes human, mouse, rat TFDB species databases and its relevance to sorghum.
Many methodological choices, such as PEG concentration, thresholds for DEMs, and the use of specific bioinformatic tools, lack justification.
In results, the statements like "Initially, without any treatment, the roots and shoots of GL98 showed significantly higher growth at 72 hours" are ambiguous. Were these measurements averaged across replicates, and how were statistical significances determined? Clarify the statistical approach (e.g., ANOVA, t-tests).
The data shifts mentioned after drought treatment lack specific numeric values or trends. Include quantifiable data or percentage changes to strengthen claims.
The description of clustering and PCA trends is generic. Incorporate more detailed interpretations of how these differences contribute to understanding drought resistance mechanisms.
The method for grouping metabolites in K-means clustering needs further detail. Specify the criteria for assigning metabolites to specific clusters and discuss biological relevance.
The Venn diagram provides visual information but lacks interpretation. Explain the significance of shared and unique DEMs across groups in Venn diagram.
Include details on how multiple testing was corrected during DEG identification (e.g., FDR, Benjamini-Hochberg).
Figures 6–9 are mentioned but lack adequate interpretation in the text. Include key findings with statistical significance of trends observed.
Enhance figures with self-explanatory captions with full forms of abbreviations and statistical thresholds are explicitly mentioned.
The discussion frequently cites other crop species, such as rice, grapes, and tea trees, without clearly linking the findings to sorghum. While comparisons are valuable, they should focus on drawing specific insights relevant to the drought resistance mechanisms of sorghum.
The study’s unique contribution, particularly in identifying 26 drought-resistance genes in sorghum, is not emphasized adequately. Consider discussing how these findings advance current knowledge or fill gaps in sorghum-specific drought research.
"3,914 metabolites" detected is impressive, but the authors need to highlight key metabolites with relevance to drought stress and provide insight into why they are important biologically.
Although the results mention integration, the discussion does not explicitly connect the transcription factors (TFs) to specific metabolic pathways (e.g., flavonoid biosynthesis) or drought-adaptive traits. This integration would significantly enhance the interpretability of the study’s conclusions.
The discussion attributes a central role to flavonoids in drought tolerance but does not substantiate this sufficiently for sorghum. Are the identified genes and metabolites directly involved in flavonoid biosynthesis in sorghum? Addressing this explicitly would strengthen the conclusions.
The specific role of KEGG and GO enriched pathways in sorghum’s drought adaptation remain unclear. Highlighting particular metabolic adjustments or physiological mechanisms would make the discussion more insightful.
The study identifies several TF families (ERF, AUX, MYB, etc.) but does not elaborate on their roles in sorghum’s drought response. Discussing their potential regulatory roles or linking them to observed phenotypic or biochemical traits would deepen the analysis.
The statement about laying a foundation for further research is too broad. Specify potential applications, such as genetic engineering targets, marker-assisted breeding, or the development of drought-resilient sorghum varieties.
There is little mention of the study’s limitations or areas for improvement. Including a brief acknowledgment of gaps, such as the need for functional validation of identified genes, would add balance to the discussion.
Specific comments:
Line 88: "the samples powder was then dissolved..." Change to "the powdered samples were then dissolved...".
Line 113: "Sequencing unfolded on an Illumina NovaSeq platform, creating an extensive set of 150 bp paired-end reads." Here replace unfolded with conducted.
Line 129: "so rghum" Typo. Should be "sorghum."
Line 204: "changes in metabolite accumulation are strictly controlled by differential gene expression" Replace "strictly controlled" with "primarily regulated".
Line 303: “In recent years, numerous histological studies have demonstrated a correlation between plant resistance and flavonoids.” Replace “numerous” with “several” for formality. Additionally, “histological studies” might not align with flavonoids unless explicitly related to tissue-level observations. Recheck.
Line 308-313: Repeated citation of Savoi et al. (2016). Consolidate references and avoid repetition.
Line 314: “Comparative transcriptome analysis of transcriptome analysis revealed...” Typographical error, correct the sentence.

Experimental design

The experimental design lacks a clear mention of the statistical packages and methods used. Please mention all this information at the end of materials and methods section under separate heading "statistical analysis".

Validity of the findings

Validity of the findings can be confirmed once the authors incorporates the suggested improvements in the manuscript.

Reviewer 2 ·

Basic reporting

‘Metabolomic and transcriptomic analysis of drought resistance mechanisms in sorghum varieties’ is a good work that helps to elucidate the mechanism of drought stress tolerance. However, it needs the following major revisions.
In the Abstract section, Line 21: Explain the abbreviations KEGG and GO at the first occurrence.
Line 26-27: Sorghum is already known for its high drought resistance. This plant is useful for us to understand the metabolic and transcriptomic mechanisms of drought tolerance. However, the need to develop drought tolerant sorghum cultivars is controversial.
The introduction section is well organized and clearly reflects the content of the research, but the purpose of the study should be expressed in clearer sentences in the last paragraph.

The discussion section is quite inadequate. You are not the first study on this subject. Instead of giving examples of plants that are very distant from sorghum, rewrite the discussion with references to plants that are closer to sorghum or to previous studies on sorghum
Highly similar to the article titled ‘Integrating transcriptomics and metabolomics to analyse quinoa (Chenopodium quinoa Willd.) responses to drought stress and rewatering’. You have been heavily inspired, especially in the writing of the results section. Reduce the similarity rate.

Experimental design

The material and method section is presented in a sufficiently explanatory manner.
In the discussion section, you wrote in Line 300-301 that you found GL98 to be drought-sensitive and GL220 to be drought-tolerant. Did you reach this finding as a result of this study? Or did you already know their drought tolerance status and chose them to explain the differences? It is a sentence that contradicts what you have written before. Please correct it.

Validity of the findings

No comment

Additional comments

no comment

Reviewer 3 ·

Basic reporting

The research described in this manuscript fits with the Aims and Scope of the journal. The purpose of the study was not clearly written in the introduction. What is the aim of this study?

Experimental design

Regarding the methods, a lot of details are lacking which would make the study difficult to replicate.
• Generally, it is difficult to understand how the experiment was set up until you finish reading the result section because sufficient explanatory information about the materials and methods is not given.
• For GL98 and GL220 varieties, a reference article stating that they are drought tolerant or sensitive varieties must be provided.
• Line 225; groups A, B, C, D should be written clearly. What do A, B, C, D mean? At the beginning all the groups should be defined. Line 235; in figure 7, it is not clear what GL98A1, A2, A3, B1,B2, B3 are. What do they represent?
• Which tissues were used for RNA isolation? It is not stated in the method that measurements were made regarding root and shoot elongation, this is a shortcoming.
• Line 75; experimental materials should be given. How did you construct drought experiments? What are your groups? What kind of observations were made after the drought experiment was not written. Also, there is no information about germination. How to set the experimental groups need to be explained. How many days-old seedlings were used to isolate RNA?
• Line 10, what is the manufacturer for the RNA isolation kit?
• For data analysis especially for principal coordinate analysis, volcano plot, heat map, GO and KEGG, more detailed information should be given. Which packages were used? Give references for these analyses.
• Which sorghum genome was used for annotation? Give information on the methods.
• The sentence ‘Twelve RNA-Seq libraries of seed tissue generated approximately 75.05 Gb of clean data’ moves to the first sentence to the section ‘Transcriptomic study of whole sorghum seedlings under drought conditions’

Validity of the findings

The impact of this study is not emphasized. I have stated my views below:
• Discussion first sentence is not understandable.
• Line 311, not ‘This study’ refer to Savoi et. al. 2016.
• In the discussion, whether the transcription factors were up- or down-regulated after drought treatment? More information about DEGs for TFs (ERF, AUX, B3, MYB, bHLH, WRKY, PHD, NAC, and bZIP ) should be provided. What is the roles of these TFs on drought resistance? More explanations need to be done.
• Line 338; more information needs to be given for a total of 26 genes related to sorghum drought resistance. It would be better to provide a separate list for these genes. This information is of critical importance to this research.
• I recommend that conclusion be rewritten to emphasize the important conclusions.
• Results and discussion about the relationship between metabolites and DEGs results are not adequately presented.
• A very similar transcriptome analysis on drought in sorghum was conducted in 2020 (given below), but, it is a shortcoming that its results were not discussed with the findings of this study.
o Abdel-Ghany, S. E., Ullah, F., Ben-Hur, A., & Reddy, A. S. (2020). Transcriptome analysis of drought-resistant and drought-sensitive sorghum (Sorghum bicolor) genotypes in response to PEG-induced drought stress. International Journal of Molecular Sciences, 21(3), 772.

Additional comments

• Line 49-57; some spelling rules were not followed. For example, gene names were not written in italics.
• Line 206; add ‘according to PCA anlaysis’
• Line 224-236, in numbers, commas should be replaced with periods.
• Line 235: ‘’Venn diagram in Figure 7 shows no common DEGs among all four groups, with 2,220, 256, 86, and 1,306 specific DEGs in each of the four comparison groups, respectively.’’ Figure 7 does not show this.
• Line 249-253 move to line 245.

Annotated reviews are not available for download in order to protect the identity of reviewers who chose to remain anonymous.

---

## Round 0.2 · Minor Revisions

Your manuscript needs some minor revisions.

Reviewer 1 ·

Basic reporting

The authors have satisfactorily addressed the majority of the suggestions raised during the previous revision. The manuscript has been substantially improved and is now suitable for acceptance for publication. However, a few minor formatting issues remain. In particular, scientific nomenclature should be consistently italicized during the proofreading stage.

Experimental design

The experimental design is technically sound.

Validity of the findings

Findings of the study seems valid.

Reviewer 2 ·

Basic reporting

After corrections, it is clear that the manuscript is better. But the similarity rate is still too high (38%), you need to lower it and what does A, B, C, D in capital letters in Figure 1 stand for? Add.

Experimental design

.

Validity of the findings

.

Reviewer 3 ·

Basic reporting

The authors have rewritten the article with the necessary edits, considering previous requests for revision. However, I need to make some edits in the following points.
1- The explanations in the section' Results: Phenotypic evaluation of GL98 and GL220 under drought stress’ are inconsistent with the graphics and photographs. For example, ‘The shoot lengths of GL98 and GL220 were similar,’ Figure 1b shows the dissimilarity. And, ‘the lengths of the roots and shoots of GL98 and GL220 decreased’. But, when Figure 1b is examined, shoot length does not decrease after 60h ve 72h stress in GL98 compared to the control. And vice versa, for GL220, shoot length increased after 60 h, but remained similar after 72h. For the root, the root length increases after 48h, 60h, and 72h of stress in GL220? The figures should be carefully explained again.
2- In the ‘RNA-seq differential expression analysis’ section, ‘A total of 33453 DEGs’ must be 3453.
3- In the ‘DEG clustering analysis’ section, ‘The expression of Cluster 1 genes increased after drought stress in GL98 and did not change between the GL220 treatment and the control.’ It looks change.
4- ‘The expression of Cluster 3 genes decreased after drought stress in GL98 and decreased less between the GL220 treatment and the control’.
5- What kind of metabolites do clusters 1, 2, 3, and 4 in Figure 8b represent?
6- In discussion, ‘After drought stress, the expression level of the sorghum SPS Sobic.009G040900.v3.2) gene increased (log2FC = 3.07)’, please check SPS gene or SPP gene. I think SPP. And gene ID should be Sobic.009G040900, not Sobic.009G040900.v3.2. Then the other sobic genes should be written in the same way. Also, gene names should be written in italics (like SPP)

Experimental design

Research question well defined, relevant & meaningful.

Validity of the findings

The explanations in the section' Results: Phenotypic evaluation of GL98 and GL220 under drought stress’ are inconsistent with the graphics and photographs. For example, ‘The shoot lengths of GL98 and GL220 were similar,’ Figure 1b shows the dissimilarity. And, ‘the lengths of the roots and shoots of GL98 and GL220 decreased’. But, when Figure 1b is examined, shoot length does not decrease after 60h ve 72h stress in GL98 compared to the control. And vice versa, for GL220, shoot length increased after 60 h, but remained similar after 72h. For the root, the root length increases after 48h, 60h, and 72h of stress in GL220? The figures should be carefully explained again.

---

## Round 0.3 · accepted · Accept

The changes made are, in my view, appropriate for the acceptance of the manuscript.